# Robust evidence for reversal in the aerosol effective climate forcing trend

Johannes Quaas[1], Hailing Jia[1], Chris Smith[2,3], Anna Lea Albright[4], Wenche Aas[5], Nicolas Bellouin[6,7], Olivier Boucher[6], Marie Doutriaux-Boucher[8], Piers M. Forster[2], Daniel Grosvenor[2], Stuart Jenkins[9], Zbigniew Klimont[3], Norman G. Loeb[10], Xioyan Ma[11], Vaishali Naik[12], Fabien Paulot[12], Philip Stier[9], Martin Wild[13], Gunnar Myhre[14], and Michael Schulz[15]

[1]Universität Leipzig, Institute for Meteorology, Leipzig, Germany
[2]University of Leeds, School of Earth and Environment, Leeds, U.K.
[3]International Institute for Applied Systems Analysis, Laxenburg, Austria
[4]Laboratoire de Météorologie Dynamique, Institut Pierre Simon Laplace, Sorbonne Université, Paris, France
[5]Norwegian Institute for Air Research, Kjeller, Norway
[6]Institut Pierre-Simon Laplace, Sorbonne Université / CNRS, Paris, France
[7]University of Reading, UK
[8]EUMETSAT, Darmstadt, Germany
[9]University of Oxford, Atmospheric, Oceanic and Planetary Physics, Oxford, U.K.
[10]NASA Langley Research Center, Hampton, USA
[11]Nanjing University of Information Science & Technology, School of Atmospheric Physics, Nanjing, China
[12]Geophysical Fluid Dynamics Laboratory, Princeton, USA
[13]ETH Zürich, Department of Environmental Systems Science, Zürich, Switzerland
[14]CICERO, Oslo, Norway
[15]Norwegian Meteorological Institute, Oslo, Norway

**Correspondence:** Johannes Quaas (johannes.quaas@uni-leipzig.de)

**Abstract.** Anthropogenic aerosols exert a cooling influence that offsets part of the greenhouse gas warming. Due to their short tropospheric lifetime of only up to several days, the aerosol forcing responds quickly to emissions. Here we present and discuss the evolution of the aerosol forcing since 2000. There are multiple lines of evidence that allow us to robustly conclude that the anthropogenic aerosol effective radiative forcing – both aerosol-radiation and aerosol-cloud interactions – has become globally less negative, i.e. that the trend in aerosol effective radiative forcing changed sign from negative to positive. Bottom-up inventories show that anthropogenic primary aerosol and aerosol precursor emissions declined in most regions of the world; observations related to aerosol burden show declining trends, in particular of the fine-mode particles that make up most of the anthropogenic aerosols; satellite retrievals of cloud droplet numbers show trends in regions with aerosol declines that are consistent with these in sign, as do observations of top-of-atmosphere radiation. Climate model results, including a revised set that is constrained by observations of the ocean heat content evolution show a consistent sign and magnitude for a positive forcing relative to 2000 due to reduced aerosol effects. This reduction leads to an acceleration of the forcing of climate change, i.e. an increase in forcing by 0.1 to 0.3 W m$^{-2}$, up to 12% of the total climate forcing in 2019 compared to 1750 according to IPCC.

# 1 Introduction

Anthropogenic pollution particles, aerosols, exert an effective radiative forcing on climate due to aerosol-radiation interactions (ERFari, also known as "aerosol direct effect", combined with the "semi-direct effect") and aerosol-cloud interactions (ERFaci, "aerosol indirect effect") (Chýlek and Coakley, 1974; Boucher et al., 2013; Forster et al., 2021; Szopa et al., 2021). ERFari occurs through the scattering and absorption of sunlight by aerosols while for ERFaci aerosols act as cloud condensation nuclei (Twomey, 1974). Both entail rapid adjustments that tend to enhance the radiative forcing. A recent assessment provided

an estimated total ERF due to aerosols (ERFaer) in the range of $-2.0$ to $-0.35\,\mathrm{W\,m^{-2}}$ (5 to 95% confidence interval; 2005 to 2015 compared to 1850, Bellouin et al., 2020b). The latest assessment report by the Intergovernmental Panel on Climate Change (IPCC) concluded that the 2019 vs. 1750 ERFaer has a best estimate of $-1.1\,\mathrm{W\,m^{-2}}$ and 5 to 95% confidence interval of $-1.7$ to $-0.4\,\mathrm{W\,m^{-2}}$ (Forster et al., 2021). This negative forcing offsets a sizeable fraction of the current $CO_2$ ERF. Throughout this manuscript, we consider ERF with 1750 as baseline, or changes in ERF over certain periods (most often from 2000 to 2019).

Forster et al. (2021) quantify a temperature increase in 2019 relative to 1750 of $+1.01°C$ due to the ERF by $CO_2$ ($+1.81°C$ considering all anthropogenic greenhouse gases), and a temperature change by $-0.50°C$ due to aerosols in that period. This implies that without the cooling effect of aerosols, the world would already have reached the $1.5°C$ temperature threshold of "dangerous" climate change as set out by the Paris agreement.

A fundamental difference between radiative forcing by aerosols and long-lived greenhouse gases is tied to their atmospheric
lifetimes: greenhouse gases have lifetimes of decades to millennia (Solomon et al., 2009), while the lifetime of tropospheric aerosols is only up to several days. Climate thus responds to long-lived greenhouse gases such as $CO_2$ largely in terms of their cumulative emissions, but to aerosols in direct link to its current emissions rate. Shorter lived greenhouse gases such as methane have an intermediate effect, whereby deep reductions in emissions can have substantial effects on temperature within a few decades (Shindell and Smith, 2019; Smith et al., 2021b; Allen et al., 2022). A further reduction in aerosol emissions – required
due to their environmental and health impacts (Lelieveld et al., 2015; Cohen et al., 2017) – thus takes out the negative aerosol forcing and leads to a warming relative to the period prior to emission reduction (Brasseur and Roeckner, 2005; Dufresne et al., 2005), an effect also known as climate penalty of air quality improvements (Ekman et al., 2020; Hong et al., 2020). Also, the importance of the aerosol forcing relative to the $CO_2$ forcing was largest in the early industrial period (Stevens, 2015). It will continue to decrease, since anthropogenic aerosol emissions will likely decrease at the global level (Myhre et al., 2015; Szopa
et al., 2021).

At what point did the aerosol forcing became substantially less negative at a scale relevant for global forcing? There are suggestions that decreasing started in the last decades in different regions, and for several regions this trend reversal has been documented (e.g., Cermak et al., 2010). The decrease stems in particular from reductions of $SO_2$ emissions from coal use in the residential sector, power plants, and industry. For other regions, evidence is lacking or more anecdotal. However, for
understanding of global climate change, it is relevant to ask to which extent the aerosol climate forcing has become less negative at the global scale.

Here we propose that aerosol trends and their effects can be best investigated in the satellite era since the turn of the century. We analyse multiple observational and model datasets to demonstrate that both ERFari and ERFaci show reduced trends since 2000 in regions which demonstrate a robust and substantial aerosol ERF trend in models.

## 2 Changes in aerosol emissions

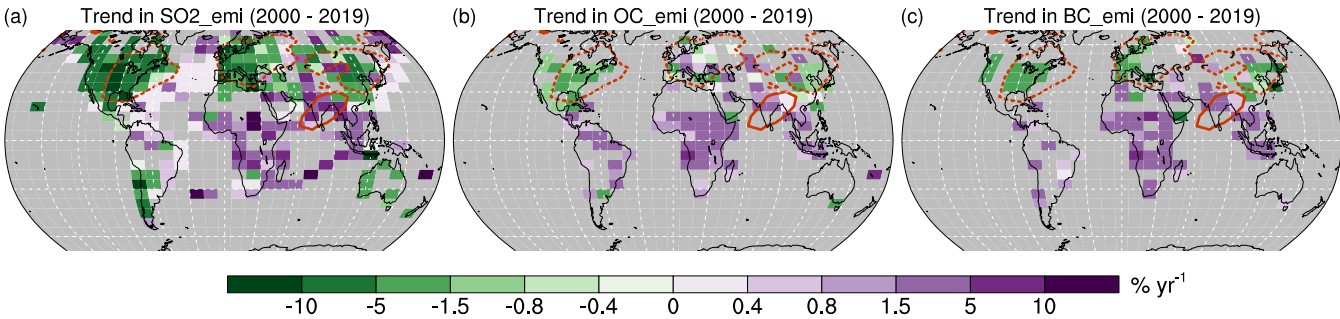

**Figure 1.** Linear trends (2000 to 2019) of (a) anthropogenic emissions in sulfur dioxide ($SO_2$) from the Community Emissions Data System (CEDS v_2021_04_21; Hoesly et al., 2018); (b) and (c) as (a), but for anthropogenic emissions in organic carbon (OC) and black carbon (BC), respectively. Regions with small absolute trends (less than $7\,\mu g\,m^{-2}\,day^{-1}\,yr^{-1}$) are masked by grey shading. Isolines enclose regions with trends in clear-sky solar ERF (see later, Fig. 4) larger than $0.05\,W\,m^{-2}\,yr^{-1}$ in absolute terms. The average values in these regions are shown in Fig. 6, listed in Table 1 and discussed in Section 7. A figure that combines the panels of Fig. 1 to 4 is provided as Supplementary material. A figure that shows the trends in absolute units is also provided as Supplementary material.

Despite substantial differences in their absolute magnitude especially at the regional level (Elguindi et al., 2020), the different emission inventories agree in general on the sign of the historical trends at regional and global levels (Granier et al., 2011; Klimont et al., 2017; Hoesly et al., 2018; Aas et al., 2019; Elguindi et al., 2020), especially over Europe and North America (Elguindi et al., 2020). A number of clear conclusions have thus been drawn in the literature for aerosol emissions in specific regions. Aerosol emissions have seen a steep increase since the beginning of the industrial period (e.g., Szopa et al., 2021). In several regions, declines after a peak are documented. An example is Europe, where since the 1980s, aerosol emissions declined strongly following air quality policies (Krüger and Graßl, 2002; Vestreng et al., 2007; Tørseth et al., 2012; Cherian et al., 2014; Crippa et al., 2016; Costa-Surós et al., 2019). A similar behaviour is documented for North America (e.g., Streets et al., 2009; Aas et al., 2019; Elguindi et al., 2020). Sulphur and nitrogen deposition over the USA, reflecting anthropogenic emissions, have been declining by between 1 and 3% $yr^{-1}$ in the period 1989–2010 (Sickles II and Shadwick, 2015). In contrast, anthropogenic aerosol emissions over China have been increasing until around 2010, and decreasing thereafter (Klimont et al., 2017; Zheng et al., 2018; Aas et al., 2019; Wang et al., 2021). The exact temporal evolution of aerosol emissions over the past 20 years especially over China was erroneously represented (a too weak decline since 2010) in some emission datasets, leading to some incompatibility of aerosols in the 6th Coupled Model Intercomparison Project (CMIP6; Eyring et al., 2016; Hoesly et al., 2018; Elguindi et al., 2020) in comparison to observations (Paulot et al., 2018; Wang et al., 2021). Aerosol emissions over

India continued to rise throughout the period 2000 to 2019 (Klimont et al., 2013; Wang et al., 2021). Over remote oceanic regions, ship emissions played a substantial, increasing role in the first part of the time period of interest (Smith et al., 2011). Since 2010, they have declined first in emission control areas (IMO, 2008) and since 2020 over much of the global oceans (IMO, 2019). This declining signal is visible also in cloud properties (Gryspeerdt et al., 2019).

Here we consider more specifically emissions from the the newest version of the Community Emissions Data System (CEDS; O'Rourke et al., 2021). The trends are shown in Fig. 1. A previous version of this dataset was used in CMIP6 and described by Hoesly et al. (2018). Sulfur emissions were mostly declining since 2000, in particular there were substantial declines over North America and Europe, continuing decreasing trends that started in the last decades of the 20th century. Also over East Asia, due to reductions after 2010, the overall trend is negative, despite the fact that in the first decade of the 21st century, emissions did

still increase. Over Southeast Asia, including India, and also over parts of Africa, sulfate precursor emissions showed increasing trends. Some shipping routes over ocean show increasing trends in this period. OC and BC emissions broadly show the same, but with more widespread increases especially over more regions in East Asia, Africa, and also South America. All considered aerosol species show increasing trends in emissions for high latitudes of both hemispheres. The updates of CEDS emissions (Elguindi et al., 2020) show that more recent evidence points to even stronger decline in $SO_2$ emissions in the second part of

the last decade and the BC and OC trends are showing a decline rather than increase, especially in China (see also Kanaya et al., 2020). These are further discussed in Elguindi et al. (2020).

## 3    Changes in aerosol abundance

The emission trends are reflected in observations of aerosol abundance. Due to their short lifetime, it is expected that regional trends in emissions are also reflected by regional trends in concentrations that are somewhat smoothed out spatially, in case

of typically prevailing wind directions mostly leewards. Trends in surface concentrations from in situ observations were found to show the expected trends in a global compilation (Collaud Coen et al., 2020) for sulfate and PM2.5, and specifically so for the declining trends over Europe (Stjern et al., 2011; Aas et al., 2019) and North America (Jongeward et al., 2016; Aas et al., 2019), and the first increasing, then decreasing behaviour over China (Zheng et al., 2018; Aas et al., 2019).

The analysis of trends from remote sensing, especially from satellite remote sensing, is challenging, because datasets may not

be homogeneous over the lifetime of a satellite instrument, due to changing instrument response and satellite orbit. However, for NASA's Earth Observation Satellites Terra and Aqua, care has been taken to avoid many of the issues that hamper satellite trend analysis such as orbital drift (Levy et al., 2013). Studies show that trends from various satellites are, at least qualitatively, consistent (Wei et al., 2019). Declining trends in aerosols in certain regions, such as over Europe (Stjern et al., 2011; Cherian et al., 2014; Li et al., 2014; Georgoulias et al., 2016; Cherian and Quaas, 2020) and over the USA (Li et al., 2014; Jongeward

et al., 2016; Cherian and Quaas, 2020), as seen from satellite analysis, have been documented earlier. The changes in aerosols over East Asia, especially China, are not monotonic over the period of interest. Rather, the trends reversed from positive (2000–2010) to negative (since 2010), and this is seen in satellite observations of aerosols (Paulot et al., 2018; Sogacheva et al., 2018; Filonchyk et al., 2019; Ma et al., 2019; Samset et al., 2019). In contrast, over Southeast Asia, especially India, aerosol

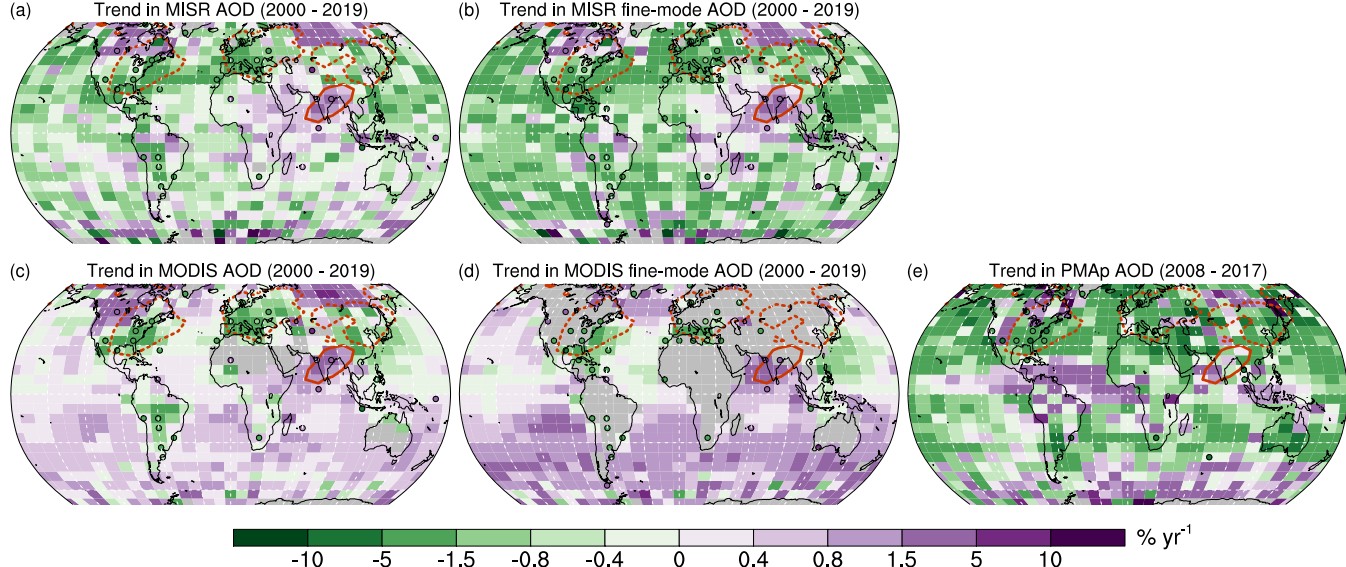

**Figure 2.** Linear trends (2000 to 2019) of (a) aerosol optical depth (AOD) as retrieved from the Multi-angle Imaging Spectroradiometer (MISR; Garay et al., 2017) on board the Terra satellite, where the coloured circles show the AOD trends from the AERONET ground-based sunphotometer network (Holben et al., 1998; Giles et al., 2019) where data since 2000 are available; (b) as (a) but for the fine-mode AOD, i.e. the AOD due to aerosols with radius smaller than 1 $\mu$m. (c) and (d) as for (a) and (b), but retrievals from the MODerate Resolution Imaging Spectroradiometer (MODIS; Levy et al., 2013) (fine-mode AOD unavailable over land) from the Terra satellite averaged (starting 2002) with MODIS retrievals from the Aqua satellite; (e) PMAp aerosol optical depth as retrieved from the Global Ozone Monitoring Experiment–2 (GOME-2) instrument on-board EUMETSAT's Metop-A satellite that is available only for 2008 to 2017. Isolines as in Fig. 1. A figure that shows the trends in absolute units is provided as Supplementary material.

retrievals from satellites show continuing increases throughout the period (Li et al., 2014; Zhao et al., 2017; Dahutia et al.,
2018; Hammer et al., 2018; Cherian and Quaas, 2020). Model-data synergy allowed to attribute these satellite-derived trends to the specific emission changes (Bauer et al., 2020; Yu et al., 2020), and to quantify the changes at between –3.1 and –1.2% yr$^{-1}$ for the different regions affected by the declines in anthropogenic aerosol emissions from 2000 to 2014 (Mortier et al., 2020). Mortier et al. (2020) further documented that climate models were able to reproduce these trends quantitatively.

    We report AOD trends from various satellite datasets on a common scale in Fig 2. It specifically shows aerosol optical depth
(AOD) and fine-mode AOD (AODf) from the MODerate Resolution Imaging Spectroradiometer (MODIS, Levy et al., 2013) instrument on board the EOS Terra and Aqua satellites, and the Multi-Angle Imaging Spectro-Radiometer (MISR, Garay et al., 2017) instrument on board the EOS Terra satellite. Also the – presumably more stable – ground-based retrievals from the AERONET network (Holben et al., 1998, 2001; Giles et al., 2019) are analysed for the stations for which the time series since 2000 are available. The regression coefficients are reported, and it is to be kept in mind that none of the quantities really follows
a straight line. Rather, the overall tendency of the noisy time change is referred to. The trends in both AOD and AODf from the different satellite instruments in the Southern hemisphere oceanic, and also in the Northern hemisphere high-latitude oceanic

regions differ – MODIS shows increases and MISR, decreases or scattered results in both quantities. As a third estimate, the EUMETSAT Polar Multi-sensor Aerosol optical properties product (PMAp, Grzegorski et al., 2021) climate data record derived using the Global Ozone Monitoring Experiment–2 (GOME-2) instrument on board EUMETSAT's Metop-A satellite are used. These are available for a shorter, 10-year period, for 2008 to 2017. In the Southern hemisphere oceanic regions, Metop-A shows very small trends for this shorter period; in the Northern hemisphere high latitude oceans, it tends to confirm the decreases shown by MISR. The increase in AOD retrieved by MODIS has been reported in previous studies. Bai et al. (2020) report an increasing trend over the 2003 to 2017 period, they propose that there might have been an increase in sea salt consistent with increasing wind speed in reanalysis. In a different conclusion, Fan et al. (2018) demonstrate that while trends in AOD from MODIS are consistent with those derived from Aeronet over land in the Northern hemisphere, the trends over Australia and Southern America are inconsistent. Also in the study by Wei et al. (2019), the MODIS trends in the Southern hemisphere were reported to show stronger positive trends than the six other satellite products they examined for the 2003 to 2010 period in the oceanic regions of the Southern hemisphere they investigated (South Atlantic and Indian oceans).

However, in the regions discussed above with pronounced trends in anthropogenic aerosol (precursor) emissions, the satellite trends show the same behaviour qualitatively in all three datasets. These trends are largely consistent with those from AERONET data (circles in Fig. 2). The decreasing trends over North America, Europe, and East Asia are clearly seen and at many grid points statistically significant (at 5% significance level according to a t-test with correction as in Santer et al., 2000), as are the increasing trends over India. It is particularly interesting to note that the trends in AODf are more consistent still in spatial extent to the changes in sulfate (precursor) emissions. These smaller particles, with radii $<1\,\mu$m, contain the bulk of the anthropogenic contribution to the aerosol (Bellouin et al., 2005; Kaufman et al., 2005; Kinne, 2019).

## 4   Changes in cloud properties

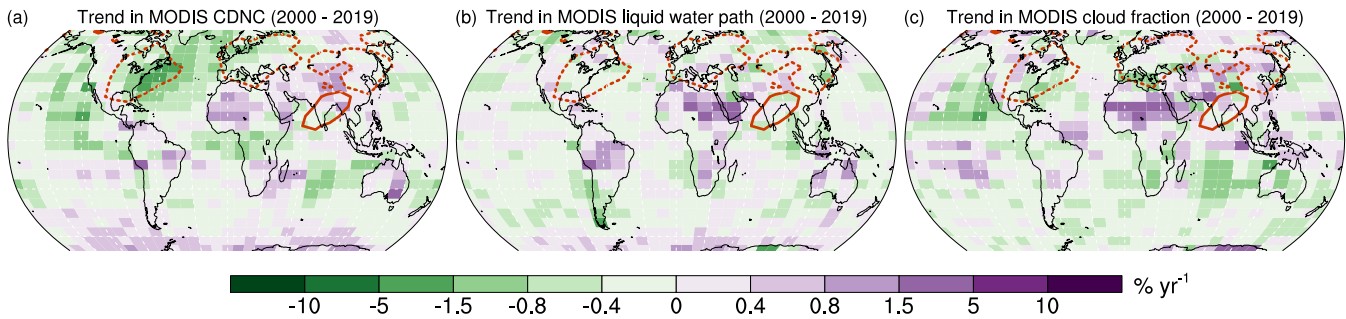

**Figure 3.** Linear trends (2000 to 2019) in cloud properties retrieved for liquid-water clouds from MODIS (Platnick et al., 2017) where cloud droplet number concentration (CDNC, panel a) and cloud liquid water path (LWP, panel b) are computed assuming adiabatic clouds (Quaas et al., 2006; Grosvenor et al., 2018); (c) liquid cloud fraction. Isolines as in Fig. 1. A figure that shows the trends in absolute units is provided as Supplementary material.

Clouds are a key determinant for variability and trends of the Earth's energy budget. Due to their large spatio-temporal variability it is not easy to distinguish long-term signals from weather noise. Clouds respond not only to aerosols, but also to global warming and interannual as well as decadal internal climate variability (Forster et al., 2021). Overall, satellite analysis

documented changes in clouds that are consistent with several of the hypotheses relevant for cloud-climate feedbacks (Norris et al., 2016), but little evidence for patterns of cloud cover or cloud-top altitude trends that would be expected due to aerosol-cloud interactions (Norris et al., 2016). The most immediate impact of aerosols is on cloud droplet number concentration (Bellouin et al., 2020b; Quaas et al., 2020). For this microphysical quantity, some clear and significant trends were identified in satellite observations for the outflow region east of East Asia (Bennartz et al., 2011), albeit the declining trends in cloud water

path and cover are not necessarily what is expected in relation to aerosol-cloud interactions (Benas et al., 2020).

The trends in satellite-derived cloud droplet number concentrations were reported previously to be consistent with the aerosol trends in several regions (McCoy et al., 2018; Cherian and Quaas, 2020). Trends in cloudiness and cloud radiative properties are, however, less conclusive possibly due to their large variability beyond the variability driven by aerosols (Norris et al., 2016; Cherian and Quaas, 2020), and also since their response to aerosols depends on cloud regime (Mülmenstädt and Feingold, 2018;

Zhang et al., 2022).

The trends in MODIS retrievals of cloud properties (Platnick et al., 2017) are shown in Fig. 3. MODIS Terra (10.30 h overpass time) is combined with MODIS Aqua (13.30 h overpass time) from 2002 onwards. Cloud droplet number concentration is derived from the MODIS retrievals as discussed in Grosvenor et al. (2018). For all three cloud quantities presented, only liquid-water clouds, as determined by the retrieval algorithm, are selected. The results confirm the qualitative consistency between

droplet number and aerosol trends. This consistency is inferred from the similarity in sign across the regions in which aerosols show spatially contingent trends attributable to the anthropogenic aerosol emissions as evident from Fig. 1 in comparison to Fig. 2, and in particular as summarized later in Table 1. Cloud droplet concentrations show declines especially over the oceans of the Northern hemisphere mid-latitudes, in particular downwind of the regions where aerosol emissions declined. The signal is much weaker over the continents, though (as also discussed by Ma et al., 2018). Cloud liquid water path (related to cloud

thickness; defined in the satellite retrievals as cloudy-sky rather than all-sky) does not show trend patterns that would be strongly related to the pattern of trends in droplet concentration. It was documented earlier that the adjustment of liquid water path to cloud droplet concentration perturbation appears to be weak in comparison to natural variability (Malavelle et al., 2017; Toll et al., 2019; Haghighatnasab et al., 2022; Chen et al., 2022). In contrast, the change in cloud fraction is broadly consistent in pattern and sign with the trends in droplet concentration. This was also suggested by satellite correlation studies (Gryspeerdt

et al., 2016; Rosenfeld et al., 2019; Christensen et al., 2020) and analysis of the response of clouds to volcanic eruptions (Chen et al., 2022). It is to be noted that cloud properties, especially outside the regions with strong aerosol changes, also respond to global warming (in particular, sea surface temperature trends under stratocumulus regions such as the Eastern Pacific) and natural variability. Thus, only the averaging over the larger contingent regions with consistent aerosol changes may allow to reduce this "noise" to infer a possible signal.

Nevertheless, the conclusion of this review of trends in cloud quantities is that cloud droplet concentrations show trends that are spatially consistent with the expectation of declining anthropogenic aerosol emissions, and that apparently also liquid-cloud

fraction trends show pattern consistent with the aerosol declines. Since these retrievals are independent of the aerosol retrievals discussed earlier, this is a strong corroboration of the earlier conclusion that satellites show a declining trend in aerosols in regions of anthropogenic emissions.

## 170  5   Changes in radiation

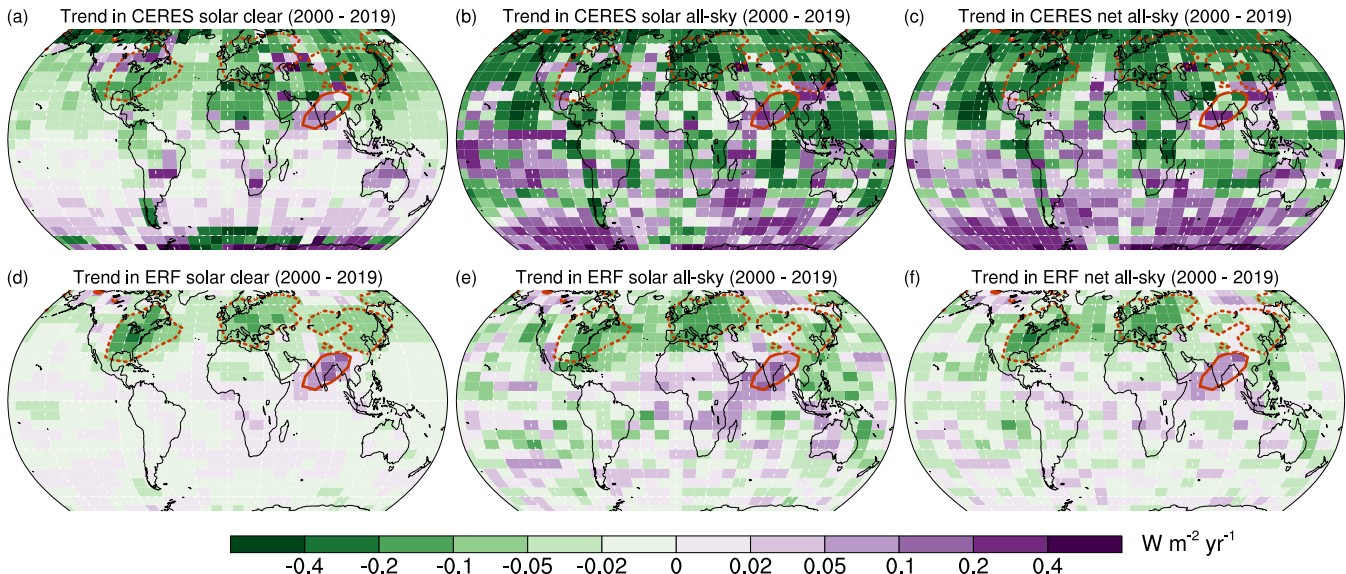

**Figure 4.** Linear trends (2000 to 2019) in (a) net broadband solar flux for clear-sky situations as retrieved from the Clouds and the Earth's Radiant Energy System (CERES) Energy Balanced and Filled (EBAF) product (Loeb et al., 2018a) from Terra, averaged for years after 2002 with retrievals also from Aqua; (b) and (c) as (a) but for all-sky solar radiation fluxes and all-sky solar plus terrestrial net fluxes, respectively; (d – f) trends in effective radiative forcings due to aerosols as computed from the dedicated RFMIP (Pincus et al., 2016) / AerChemMIP (Collins et al., 2017) for which output was available from the CanESM5 (Swart et al., 2019), GISS-E2-1-G (Kelley et al., 2020), HadGEM3-GC31-LL (Andrews et al., 2019), IPSL-CM6A-LR (Boucher et al., 2020), MIROC6 (Hajima et al., 2020), NOAA-GFDL (Held et al., 2019), and NorESM2-LM (Seland et al., 2020) of which the ensemble average is shown for (d) clear sky solar, (e) all-sky solar and (f) all sky net (solar plus terrestrial) spectra. Signs are inverted for consistency with the CERES results (negative trends in ERFaer mean decreases in absolute magnitude). Isolines enclose regions with trends in clear-sky solar ERF (panel d) larger than $0.05\,\mathrm{W\,m^{-2}\,yr^{-1}}$ in absolute terms. The average values in these regions as listed in Table 1 and discussed in Section 7. Net fluxes, defined as positive if downward, are plotted, consistently in the simulations and the satellite retrievals.

Changes in net top-of-atmosphere radiation fluxes in a period correspond to the changes in ERF in that period, but also include the signal of natural variability and of feedbacks to changing climate. Previous analysis of model simulations suggested that between 2000 and 2015, ERFaer was reduced in absolute magnitude, i.e. increased (became less negative), by about $0.003\,\mathrm{W\,m^{-2}\,yr^{-1}}$ at a global scale (Myhre et al., 2017), mainly over the Northern hemisphere mid-latitudes, especially

over North America, the North Atlantic ocean, Europe and adjacent Asia. In Fig. 4, the trends in ERFaer as simulated by models contributing to CMIP6 are analysed. This makes use of the dedicated simulations of the Radiative Forcing Model Intercomparison Project (RFMIP, Pincus et al., 2016) that trace the aerosol ERF over time (the piClim-histaer simulations). For seven Earth System Models, the relevant diagnostics for these simulations were submitted, namely for the Canadian Centre for Climate Modelling and Analysis (CCMA, Swart et al., 2019), for the US National Aeronautics and Space Administration

Goddard Institute for Space Studies Earth system model (GISS-E2-1-G, Kelley et al., 2020), for the UK Hadley Centre Global Environment Model (HadGEM3-GC31-LL, Andrews et al., 2019), for the Institut Pierre Simon Laplace Climate Model (IPSL-CM6A-LR, Boucher et al., 2020), for the Model for Interdisciplinary Research on Climate (MIROC6, Hajima et al., 2020), for the National Oceanic and Atmospheric Administration / Geophysical Fluid Dynamics Laboratory Climate Model (CM4, Held et al., 2019), and for the Norwegian Earth System Model (NorESM2-LM, Seland et al., 2020). The ensemble average of these

models is considered. The results show that the pattern in the clear sky solar ERFaer trends is closely related to the pattern in the trends in sulfate precursors. It reflects the strong declines in the main source areas over North America, Europe, and East Asia, along with the increases over India and surrounding areas. The patterns in all-sky ERFaer, i.e. including the cloud effects, both in solar and terrestrial spectra, are noisier but also show trends that are consistent with the pattern seen for the clear-sky, solar ERFaer and in aerosols and droplet concentrations also in the observations. It is interesting to assess the relative

importance of aerosols to other agents that are examined in the RFMIP piClim-histall simulations (Supplementary Fig. S5). In the solar spectrum, as expected, clearly the aerosol signal dominates, whereas in the terrestrial spectrum, the additional signal by the increase in greenhouse gases is seen. Supplementary Fig. S6 to S8 examine the differences across models in the simulated aerosol ERF trends. Despite differences in particular in the absolute magnitude, the pattern identified in the multi-model mean is qualitatively similar in all seven individual models. Similarly, for individual ensemble members of a selected ESM

(Supplementary Fig. S9 to S11) the pattern is robustly simulated.

In the multi-model mean, global mean changes show a decline of the clear-sky, solar ERFaer by $0.0117\,\mathrm{W\,m^{-2}\,yr^{-1}}$, of the all-sky solar ERFaer a decline by $0.0172\,\mathrm{W\,m^{-2}\,yr^{-1}}$ and of the all-sky terrestrial ERFaer a (compensating) increase by $0.0013\,\mathrm{W\,m^{-2}\,yr^{-1}}$. The integral, net decline over the 20-year period according to these models thus was $0.32\,\mathrm{W\,m^{-2}}$.

This result can be compared to the assessment by IPCC AR6 (Forster et al., 2021). Their assessment is based on multiple lines

of evidence that are incorporated in an emulator ensemble simulation. The time series of the diagnosed ERFaer is available via the IPCC web site and at https://doi.org/10.5281/zenodo.5705391. Computing the linear trend between 2000 and 2019 on the basis of the emulator ensemble yields an increase by $+0.0145\,\mathrm{W\,m^{-2}\,yr^{-1}}$ between 2000 and 2019 (5 to 95% confidence interval of $+0.0068$ to $+0.0253$), i.e. by $+0.29$ ($+0.14$ to $+0.51$) $\mathrm{W\,m^{-2}}$ over the full period (Gulev et al., 2021; Forster et al., 2021).

The ERFaer may be inferred from the Earth radiation budget which is measurable at the top of the atmosphere. Several studies have investigated the retrievals of this quantity from the Clouds and the Earth's Radiant Energy System (CERES) Energy instrument that is also on the EOS Terra and Aqua satellites. CERES shows patterns for clear-sky broadband radiation that are consistent with the aerosol spatio-temporal changes (Loeb et al., 2018b; Paulot et al., 2018; Loeb et al., 2021b). Further, Loeb et al. (2021a) document an increase in the Earth's energy imbalance, seen in both Earth radiation budget satellite observations

and ocean heat content. They find this to be due to a strongly decreasing trend in reflected solar radiation, which they attribute to decreased reflection by clouds and sea ice, and a declining trend in emitted terrestrial radiation due to increases in greenhouse gases and water vapour. Using partial radiative perturbation analysis, Loeb et al. (2021a) attribute the trend in solar radiation mostly to changes in clouds, with a very small contribution only due to the effect by aerosol-radiation interactions. This is also a result of a new study by Jenkins et al. (2022). CERES observations were also analysed by Raghuraman et al. (2021). They find for the period 2001 to 2020 an increasing trend by $0.038 \pm 0.024 \, \mathrm{W \, m^{-2} \, yr^{-1}}$. They attribute about one third of this trend to the reduction in aerosol ERF.

Kramer et al. (2021) disentangle the trends in satellite-retrieved radiation fluxes using radiative kernels, notably isolating the impact of radiative forcings. They quantify the change in absorbed solar radiation over the 2003 to 2018 period at $0.044 \pm 0.02 \, \mathrm{W \, m^{-2} \, yr^{-1}}$. Singling out the instantaneous radiative forcing in the solar spectrum, they obtain a change of $0.006 \pm 0.003 \, \mathrm{W \, m^{-2} \, yr^{-1}}$ which they largely attribute to aerosol changes. Paulot et al. (2018) constrained the radiative forcing due to aerosol-radiation interactions (aerosol direct effect) in the GFDL climate model and obtained an almost negligible trend in ERFaer of $0.0002 \, \mathrm{W \, m^{-2} \, yr^{-1}}$. Their study, however, considered the period from 2001 to 2015 only and thus a time when then increasing emissions over China were much more relevant.

Bellouin et al. (2020a) used the Copernicus reanalysis of atmospheric composition, which assimilates MODIS AODs, to estimate RFaer and found statistically significant decreasing (less negative) trends over North and South America, Europe, and China, and an increasing (more negative) trend over India for the period 2003 to 2017. Their globally averaged trend in RFaer is $0.00 \, \mathrm{W \, m^{-2} \, yr^{-1}}$, but limitations in their estimate may imply that the real trend is positive.

Surface measurements of radiation also show increasing trends over large regions (Wild, 2009, 2012; Cherian et al., 2014; Hatzianastassiou et al., 2020). Trends in aerosol effects become particularly apparent in surface radiation records under cloud-free conditions. Such records indicate in Europe increasing clear-sky surface solar radiation and thus decreasing aerosol effects throughout the 2000s with some tendency for saturation (leveling off) after 2010 (Manara et al., 2016; Wild et al., 2021). Surface radiation records in China suggest a trend reversal in clear sky surface solar radiation from decrease to increase in the late 2000s (Yang et al., 2019), in line with anthropogenic aerosol emission trends (Section 2). It has been shown for Europe (Pfeifroth et al., 2018) and China (Wang et al., 2019) that solar radiation consistently increases in both surface and satellite observations.

The CERES data are also shown in Fig. 4. The clear-sky solar radiation changes in the areas where the models show decreases in the clear-sky solar ERFaer, with a pattern consistent in sign and magnitude with the model results. For all-sky radiation, the data show larger trends and also much more noise in the patterns. However, the sign of the changes in the regions where an aerosol signal is expected is consistent between the models and the data. These results are consistent with what was documented in the literature before (see above). A quantiative comparison is provided later in Section 7.

## 6 Ocean heat uptake and surface temperatures as constraints for the simulated ERFaer evolution

The temporal evolution of observed climate change – specifically, surface temperature changes and their pattern – has been proposed as a constraint on the magnitude of the aerosol effective radiative forcing (Ekman, 2014; Rotstayn et al., 2015; Stevens, 2015; Kretzschmar et al., 2017; Aas et al., 2019; Albright et al., 2021; Smith and Forster, 2021). Increasingly it is now recognized that the ocean heat uptake is of overwhelming interest for monitoring the Earth energy imbalance (von Schuckmann et al., 2016; Palmer, 2017; Allison et al., 2020; Forster et al., 2021), since it is a non-volatile indicator of climate change.

Based on this, Smith et al. (2021a) constrained the aerosol ERF from the CMIP6 models by considering the ocean heat uptake from observations between 1971 and 2018, in addition to observations of surface temperature. In Smith et al. (2021a) a 100,000 member ensemble of time series of historical aerosol forcing was generated from emissions of BC, OC and $SO_2$ using simple formulas calibrated to CMIP6 models, considering ERFari and ERFaci separately, with 1850–2010 ERFari and ERFaci constrained to the distribution of Bellouin et al. (2020b) in the prior. Weights were assigned to each ensemble member based on how closely historical surface temperature and ocean heat content change were simulated compared to observations when the aerosol forcing was combined with other historical forcings in a two-layer energy balance model (Geoffroy et al., 2013) to generate a posterior distribution of historical aerosol forcing. This study assessed the ERFaer between 1750 and 2019 at –0.9 $\mathrm{W\,m^{-2}}$ and suggested a slightly positive trend between 1980 and 2014 of 0.0025 $\mathrm{W\,m^{-2}\,yr^{-1}}$. The method of Smith et al. (2021a) is applied to assess the trend in aerosol ERF between 2000 and 2019 (Fig. 5) focusing on the emissions trends during this period. This yields a constrained trend of 0.0114 (–0.003 to 0.0274) $\mathrm{W\,m^{-1}\,yr^{-1}}$, much stronger than the one considering the longer period (5 to 95% confidence interval in brackets). The integral change in the 2000 – 2019 period in ERFaer is thus 0.23 $\mathrm{W\,m^{-2}}$ in the best estimate, very close to the suggestion by the analysed models.

Albright et al. (2021) explore bounds on ERFaer using a Bayesian model of aerosol forcing and Earth's multi-timescale temperature response to radiative forcing, finding a best estimate present-day lower bound of –1.3 $\mathrm{W\,m^{-2}}$. In Fig. 5, their method is applied to the period investigated here, from 2000 to 2019. Their baseline estimate yields a mean trend of 0.0047 $\mathrm{W\,m^{-2}\,yr^{-1}}$ (5 to 95% confidence interval of –0.000912 to 0.0106 $\mathrm{W\,m^{-2}\,yr^{-1}}$). The best estimate of the change in ERFaer for the 20-year-period is thus 0.094 $\mathrm{W\,m^{-2}}$ (5 to 95% confidence interval of –0.018 to 0.21 $\mathrm{W\,m^{-2}}$). Prescribing internal climate variability that is a factor of five larger than the CMIP6 mean and assuming large, correlated errors in global temperature observations yields a fifth-percentile ERFaer lower bound of -1.8 $\mathrm{W\,m^{-2}}$ and a mean estimate of the change in ERFaer for the 20-year-period of 0.16 $\mathrm{W\,m^{-2}}$ (5 to 95% confidence interval of 0.04 to 0.32 $\mathrm{W\,m^{-2}}$, see 'increased variance' in Fig. 5b).

Albright et al. (2021) caution that ocean heat content data do not, at present, offer robust additional constraints on ERFaer. Biases in coupled climate models towards too positive radiative feedbacks since about 1980, compared to radiative feedbacks in models forced by historical surface temperatures (e.g., Zhou et al., 2016), suggest that emulators trained on coupled models yield inferences of ERFaer that are biased low when attempting to fit to recent planetary heat uptake (see Sec. 4d in Albright et al., 2021)). That is, the discrepancy between the observed planetary heat uptake and net top-of-atmosphere radiative imbalance in coupled models can be thought of as a 'ghost forcing' that can either be attributed to more negative radiative feedbacks or more negative radiative forcing (e.g., more negative ERFaer). They conclude that constraining ERFaer with ocean heat

content data requires independent observational constraints on the true radiative damping and a better understanding whether recent SST patterns that cause more negative feedbacks are forced or unforced (Andrews et al., 2018; Sherwood et al., 2020).

Jenkins et al. (2022) also suggests that $0.2\,\mathrm{W\,m^{-2}}$ is a plausible best-estimate ERF change for the considered period, based on analysis of satellite observations and energy balance of global temperatures. However, they also conclude that large variability signals implies that a very weak trend change cannot be ruled out either.

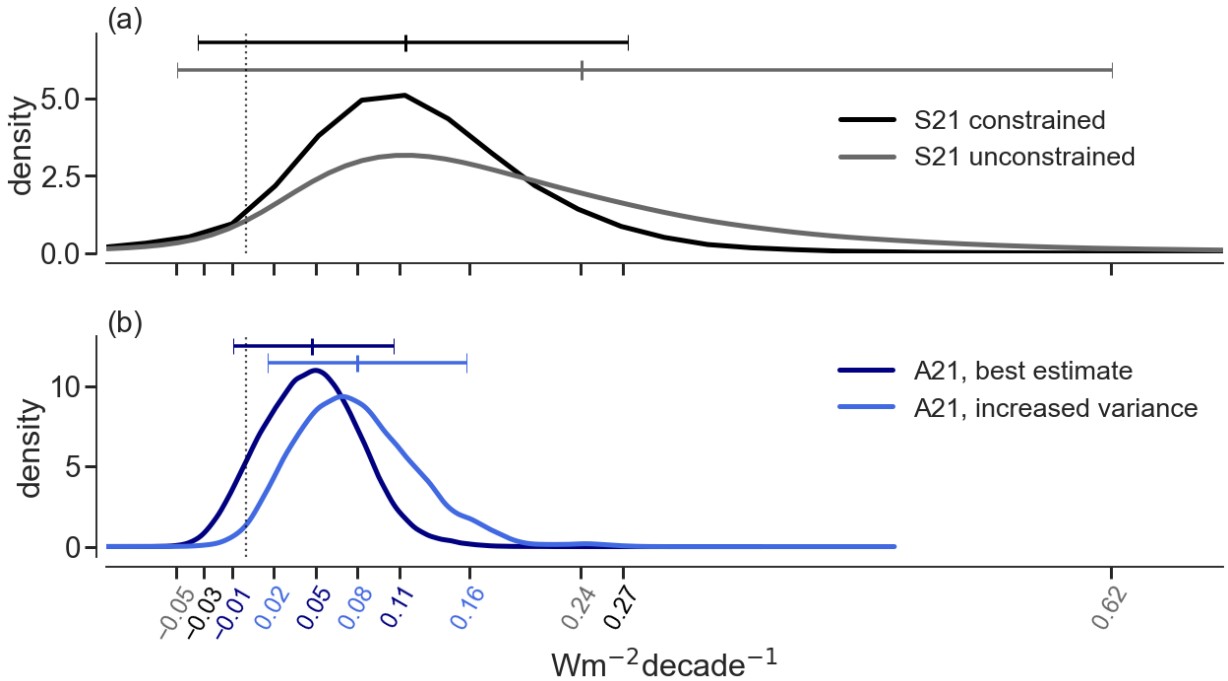

**Figure 5.** Assessment of the linear trend in ERFaer between 2000 and 2019. (a) as in Smith et al. (2021a, their Fig. 7; abbreviated as S21 in the labels), and (b) as in Albright et al. (2021, labelled A21). The constraint is as in the cited studies, but applied to the period 2000 to 2019. Increased variance in Albright et al. (2021) corresponds to a scenario prescribing internal climate variability that is a factor of five larger than the CMIP6 mean and assuming large, correlated errors in global temperature observations, yielding a fifth-percentile ERFaer lower bound of $-1.8\,\mathrm{W\,m^{-2}}$. The labels along the x-axis correspond to 5 and 95% percentiles, as well as mean, of the distribution of the curves in the corresponding colour.

## 7 Discussion and conclusions

The trends of aerosols, clouds and radiation in the observations are subject not only to changes in anthropogenic aerosol emissions, but also to other influencing factors. These include changes in natural aerosol emissions, which remain poorly constrained and contribute a substantial fraction of total observed AOD, interannual variability, and responses to greenhouse-

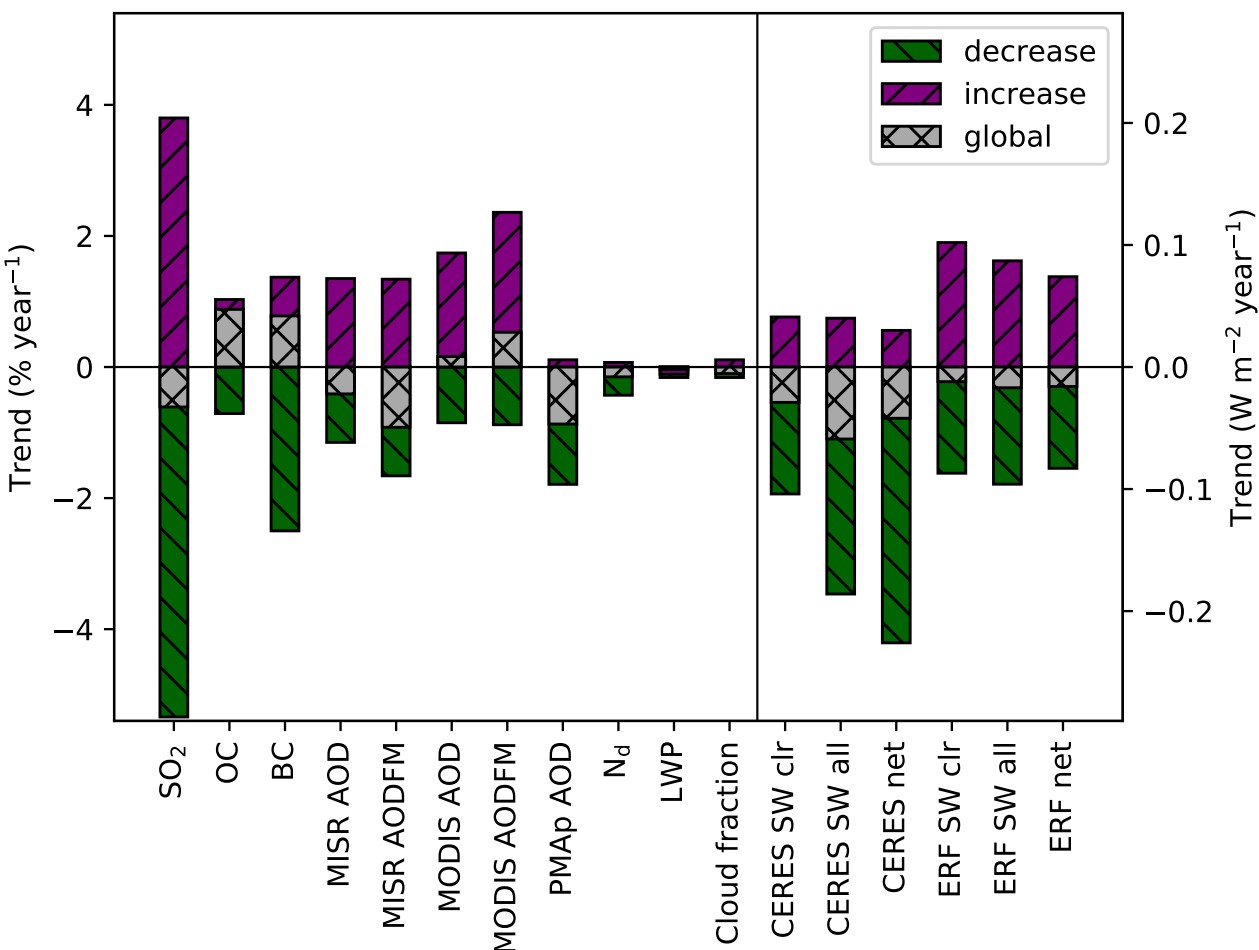

**Figure 6.** Mean values derived from the maps in Fig. 1 to 4, averaged over (green/downward hatching) the regions with substantial negative trends (defined as larger than $0.05\,\mathrm{W\,m^{-2}\,yr^{-1}}$ in absolute terms; isolines in Fig. 4) in ERF clear sky, solar from RFMIP (Fig. 4d) and (purple/upward hatching) substantial positive trends. The grey bar (crosses) is the global mean between 60°S and 60°N. The regions with negative trends cover 7.3% of the Earth surface, the ones with positive trends, 1.1%. AODFM is retrieved only over oceans, and the PMAp time series only spans ten years from 2008 to 2017. From left to right: Trends in emissions of $SO_2$, OC and BC, in AOD and AODFM from MISR, MODIS and Metop-A, in cloud droplet concentration $N_\mathrm{d}$, LWP and cloud fraction from MODIS. All these are provided in units of % $\mathrm{yr^{-1}}$ and refer to the left axis. Trends in CERES retrievals of solar clear-sky, solar all-sky and net all-sky radiation, as well as in model-derived ERF for solar clear-sky, solar all-sky, and net all-sky are shown in units of $\mathrm{W\,m^{-2}\,yr^{-1}}$ (right axis) and are inverted in sign: negative trends here are for reduction in magnitude (less negative fluxes / effective forcings).

gas-induced global warming; aerosol-cloud interactions may also be altered in a changing climate (Zhou et al., 2021; Murray-
Watson and Gryspeerdt, 2022).

**Table 1.**

| (a) SO₂ emissions (% yr⁻¹) | | | (b) OC emissions (% yr⁻¹) | | | (c) BC emissions (% yr⁻¹) | | |
|---|---|---|---|---|---|---|---|---|
| Decreases | Increases | Global | Decreases | Increases | Global | Decreases | Increases | Global |
| -5.34 | +3.80 | -0.61 | -0.71 | +1.03 | +0.88 | -2.50 | +1.37 | +0.78 |
| **(d) MISR AOD (% yr⁻¹)** | | | **(e) MISR AODFM (% yr⁻¹)** | | | | | |
| -1.15 | +1.35 | -0.41 | -1.66 | +1.34 | -0.92 | | | |
| **(f) MODIS AOD (% yr⁻¹)** | | | **(g) MODIS AODFM (% yr⁻¹)** | | | **(h) PMAp AOD (% yr⁻¹)** | | |
| -0.85 | +1.74 | +0.16 | -0.88 | +2.36 | +0.53 | -1.79 | +0.11 | -0.87 |
| **(i) $N_\mathrm{d}$ (% yr⁻¹)** | | | **(j) LWP (% yr⁻¹)** | | | **(k) Cloud fraction (% yr⁻¹)** | | |
| -0.43 | +0.07 | -0.15 | -0.16 | -0.11 | -0.04 | -0.16 | +0.11 | -0.10 |
| **(l) CERES rsutcs ($\mathrm{W\,m^{-2}\,yr^{-1}}$)** | | | **(m) CERES rsut ($\mathrm{W\,m^{-2}\,yr^{-1}}$)** | | | **(n) CERES net ($\mathrm{W\,m^{-2}\,yr^{-1}}$)** | | |
| -0.104 | +0.041 | -0.029 | -0.186 | +0.040 | -0.059 | -0.226 | +0.030 | -0.042 |
| **(o) ERF SW clr ($\mathrm{W\,m^{-2}\,yr^{-1}}$)** | | | **(p) ERF SW ($\mathrm{W\,m^{-2}\,yr^{-1}}$)** | | | **(q) ERF net ($\mathrm{W\,m^{-2}\,yr^{-1}}$)** | | |
| -0.087 | +0.102 | -0.012 | -0.096 | +0.087 | -0.017 | -0.083 | +0.074 | -0.016 |

**Table 1.** Values corresponding to Fig. 6. The trends in absolute units are reported in Supplementary Table S1.

| | |
|---|---|
| IPCC AR6 | +0.29 (+0.14 to +0.51) $\mathrm{W\,m^{-2}}$ |
| Method of Smith et al. (2021), constraint from ocean heat uptake | +0.23 (-0.05 to 0.55) $\mathrm{W\,m^{-2}}$ |
| Method of Albright et al. (2021), constraint from surface temperature changes | +0.094 (-0.02 to 0.21) $\mathrm{W\,m^{-2}}$ |
| RFMIP models (Fig. 4) | +0.32 $\mathrm{W\,m^{-2}}$ |
| Kramer et al. (2021) | +0.12 $\mathrm{W\,m^{-2}}$ |
| Raghuraman et al. (2021) | +0.24±0.20 $\mathrm{W\,m^{-2}}$ |

**Table 2.** Estimates of ERFaer change between 2000 and 2019. The 5 to 95% uncertainty ranges are provided. Kramer et al. (2021) assess RF due to aerosols and use the period 2003 to 2018; Raghuraman et al. (2021) the period 2001 to 2019.

Natural aerosol emissions, especially of dust, are highly variable and impact the distribution of AOD in specific regions (Chin et al., 2014). Natural aerosol emissions may respond to increasing temperatures (Yli-Juuti et al., 2021). Also volcanic aerosol emissions, both from eruptions and from degassing, are an important contribution in particular to atmospheric sulfate aerosol. However, in satellite retrievals for the 2005 to 2016 period, no strong trends across volcanoes have been observed (Carn et al., 2017). Fires may emit large amounts of aerosols. This was in particular the case for the Australian bush fires in 2020 (Boer et al., 2020; Heinold et al., 2021). Whether there were substantial trends in fire aerosol emissions in recent decades is not quite clear (Doerr et al., 2016), even if burned area decreased in many regions (Andela et al., 2017). However, global warming increases the risk of fire (van Oldenborgh et al., 2021). More broadly, biomass burning aerosols are typically considered separately from the anthropogenic emissions presented in Fig. 1. In particular in the northern hemisphere high

latitudes, increasing emissions in biomass burning occurred in the period of interest (van et al., 2017) and likely explain the increases in aerosol abundance (Fig. 2) seen in these regions.

Sea salt aerosols, that are a function of near-ocean-surface wind speed, are subject to variability, both forced and unforced, albeit to a lesser extent than dust (Stier et al., 2006). To the extent that the MODIS, rather than MISR, AOD and AODFM trends above the southern hemisphere oceans are right, such variability in sea salt aerosol may cause the increasing trends (Struthers et al., 2013). Trends in long-range transport of aerosols could be another reason for such increases. However, the satellite retrievals are particularly uncertain in this region, due to the large zenith angles and large cloud cover that both hamper aerosol retrievals.

Towards the end of the time series investigated here, there were specific effects due to the COVID-19 pandemic in 2020 (Forster et al., 2020; Gettelman et al., 2021; Fiedler et al., 2021). For this reason, and due to the particularly large fire activity, the end year of the data analysed here was chosen as 2019.

The different quantities investigated here are not independent. The climate models are driven by the emissions. In turn, the emissions inventories consider satellite retrievals for some of their aspects such as fires and shipping. The satellite retrievals of cloud properties and radiation are not linked to the other quantities but are more noisy in their results. Cloud properties respond to variability in climate dynamics, both forced and unforced, beyond the impact of anthropogenic aerosols. Norris et al. (2016) document global patterns of changes in cloud coverage and cloud albedo that show a reduction in cloud cover and albedo in the mid-latitudes and an increase in the Tropics from the 1980s to the first decade of the 21st century, and that are consistent with the expectations due to cloud responses to global warming.

We now turn to discussing quantifying the changes in aerosols, clouds, and radiation. For this, the regions with clear trends in aerosols are identified by subjectively choosing the regions in which the ERF simulated by the CMIP6 models (Fig. 4) exceeds $\pm 0.05\,\mathrm{W\,m^{-2}\,yr^{-1}}$ for the solar, clear sky component. Regions with increasing and decreasing ERF are distinguished. Table 1 summarizes all quantities analysed in Fig. 1 to 4. In the regions with declining clear-sky solar ERFaer, in particular $SO_2$ emissions decreased strongly, but also OC and BC emissions decrease according to the inventory of Hoesly et al. (2018). In regions with increasing clear-sky solar ERFaer, emissions of all three species increased. Both MODIS and MISR show corresponding declining trends in column-aerosol metrics for the regions with aerosol emission reductions, and increasing trends where aerosols increased. The numbers are much larger for MISR than for MODIS for the declining-trend regions (almost a factor of 2 larger in case of AODFM). Also in the global average, the MODIS-derived AOD and AODFM trends are positive, in consistent with the result from MISR and Metop-A, and also inconsistent with the trends in clouds and radiation. This is due to the aerosol increases in MODIS over much of the oceans, especially in the southern hemisphere. The reasons are debated in the literature (see above). In contrast, MODIS-retrieved cloud droplet number, liquid water path and cloud fraction in these regions all increase (decrease) where aerosol emissions increase (decrease). Globally, all three quantities decrease. Droplet number concentrations change by a rate that is a factor of 2 (compared to MODIS AOD) to 4 (compared to MISR AODFM) less than for aerosol optical depth, highlighting that there is not a 1:1 relationship of droplet number and aerosol measured as AOD (e.g., Quaas et al., 2020; Jia et al., 2022). Since the regions with increasing aerosol are more limited in extent, the absolute numbers are expected to be more uncertain. There is only a small LWP response that is inconsistent in

sign between regions with increasing and decreasing aerosol emissions. LWP does not just respond to $N_d$ perturbations, but also to global warming (with an expected increase in LWP on average; e.g., Norris et al., 2016). However, the fact that there is little LWP trend where $N_d$ trends are substantial is consistent with other observations-based assessments (Malavelle et al., 2017; Toll et al., 2019). Cloud fraction, in turn, appears to show decreasing trends in the regions with anthropogenic aerosol decreases, and increasing trends where aerosol increases. Although it is also a function of other drivers, this could be a hint at

a systematic (positive, i.e. negative in terms of forcing) aerosol effect on cloud fraction as also documented in earlier statistical studies (Gryspeerdt et al., 2016; Rosenfeld et al., 2019; Christensen et al., 2020; Chen et al., 2022). It is to be noted that the spatial consistency of the trends in cloudiness are not a proof of causality. Consistent with these results, CERES shows decreasing trends in top-of-atmosphere radiation budget. The changes in net radiation retrieved by CERES are expected to reflect the trends in ERF, but also natural variability and feedbacks to climate change. The numbers are stronger for all-sky

than for clear-sky, indicating a comparatively strong contribution by aerosol-cloud interactions (Forster et al., 2021; Loeb et al., 2021a). The numbers are consistent in sign with what the CMIP6 models suggest for changes in ERFaer, although more negative (less positive where aerosols increase) in particular when the cloud effects are included. The global average values similarly show stronger declines than CMIP6 but are also consistent in sign.

In conclusion, there are clear, robust and consistent signals for net declining anthropogenic aerosol influence on climate in the

period since 2000, i.e. the period for which high-quality satellite retrievals of all relevant quantities are available. The regions in which aerosol emissions declined (in particular North America, Europe and East Asia) dominate over regions with increasing trends. The summary of the results in terms of aerosol effective climate forcings are listed in Table 2. This demonstrates consistency of the findings of this study to previous ones. The overall climate-relevant signal is a decline in negative aerosol effective radiative forcing by about 0.1 to 0.3 $\mathrm{W\,m^{-2}}$, i.e. between 15 and 50% of the 0.6 $\mathrm{W\,m^{-2}}$ increase in $CO_2$ ERF (Forster

et al., 2021) in the same time period. This signal will very likely continue in the future, increasing the urgency for strong measures on reducing greenhouse gas emissions (McKenna et al., 2021).

*Data availability.* The MODIS cloud products MYD08_D3 from Aqua and MOD08_D3 from Terra were used in this study from the Atmosphere Archive and Distribu- tion System (LAADS) Distributed Active Archive Center (DAAC), https://ladsweb.nascom.nasa.gov/. MISR data were obtained from the NASA Langley Research Center Atmospheric Science Data Center (https://opendap.larc.nasa.gov/opendap/

MISR/MIL3YAEN.004). CERES data were obtained from https://ceres.larc.nasa.gov/data/. The Metop-A data are available as PMAp Climate Data Record (CDR) at https://doi.org/10.15770/EUM_SEC_CLM_0053. AERONET data were used from https://aeronet.gsfc.nasa.gov/data_push/AOT_Level2_Monthly.tar.gz. RFMIP model output is available from the Earth System Grid Federation (ESGF).

*Author contributions.* The fundamentals stem from active discussions with all authors. J.Q. coordinated the study and led the writing of the manuscript with significant contributions from all authors. H.J. created Fig. 1 to 4, including processing the data, and Table 1. with substantial

help from C.S. for the RFMIP models and M.D.-B. for the Metop-A data. A.-L.A. and C.S. prepared Fig. 5 and the data processed for it.

*Competing interests.* The authors declare there are no competing interests.

## 8   Acknowledgments

This work stems from discussions and work in the EU Horizon2020 projects FORCES (GA no. 821205) and CONSTRAIN (GA no. 820829). H.J. and J.Q. further acknowledge support by the German Research Foundation (Joint call between National

Science Foundation of China and Deutsche Forschungsgemeinschaft, DFG, GZ QU 311/28-1, project "CloudTrend"). PS acknowledges funding from the European Research Council (ERC) project RECAP under the European Union's Horizon 2020 research and innovation program with grant agreement 724602.

We would like to thank Annica Ekman (Stockholm University), Bjorn Stevens (Max Planck Institute for Meteorology, Hamburg), and Songmiao Fan (GFDL Princeton) for insightful comments.

We thank all data producers to make their data available. We acknowledge the World Climate Research Programme, which, through its Working Group on Coupled Modelling, coordinated and promoted CMIP6. We thank the climate modeling groups for producing and making available their model output, and the multiple funding agencies who support CMIP6 and ESGF. We would also like to thank Michael Diamond and two anonymous reviewers for constructive and helpful comments. We further acknowledge the Open Access Publishing Fund of Leipzig University supported by the German Research Foundation within

the program Open Access Publication Funding.

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
