# Peer review of "Robust evidence for reversal in the aerosol effective climate forcing trend"

_Atmospheric Chemistry and Physics, 2022_

## Author Comment (AC1)

The authors lay out a series of trends in anthropogenic aerosol and precursor emissions, column aerosol burdens, aerosol-influenced cloud properties, and top-of-atmosphere radiation that together provide consistent evidence of a reversal in the aerosol radiative forcing trend from more negative values over the twentieth century to less negative (positive trend) over the twenty-first century. This is mainly driven by trends in North America, Europe, and eastern Asia (especially since ~2010) and is somewhat offset by trends in south Asia. The manuscript is a useful review of the trends and related literature and is particularly helpful in putting everything together in one place (e.g., Table 1 and the Supplemental Figure). I believe that some further reporting of the regional breakdowns and of absolute (in addition to relative) changes would strengthen the paper. I recommend prompt publication of a suitably revised manuscript. -MD

*On behalf of the co-authors I would like to thank Michael Diamond for the thorough review of our manuscript.*

General comments

A. Relative versus absolute trends. I understand why the authors chose to report all trends (except for radiative fluxes) in relative, rather than absolute, units. Unfortunately, this choice would make it a bit difficult for someone not already familiar with the spatial pattern of aerosol burden to see the bigger picture. For instance, in Figure 2, one might think the global average trend is of opposite sign between MODIS and MISR just based on the maps shown, although if you were to take the global average, my impression is that both would show a decrease in AOD(f). Perhaps an additional supplemental figure, like the one already included but with absolute units, would be helpful?

*This is indeed a very good suggestion and in the revised version a second variant of Fig. 1-3 is included as supplementary material that shows the trends in absolute units.*

B. Regional breakdown. Table 1 has a nice breakdown of the increasing versus decreasing areas, although I would be interested in seeing a finer regional breakdown (i.e., North America, Europe, east Asia, south Asia, all other). Waterfall plots showing the global change between 2000 and 2019 and the components related to each region for some key variables (e.g., AOD, CDNC, rsutcs) could be really nice, although even just another table or an expansion of Table 1 would suffice.

*The reviewer raises a good point here. We examined thoroughly the possibility to seek reliable results also for smaller regions, but it is, as the reviewer suspected, rather difficult and noisy. It will certainly be of interest to follow this up more closely in future studies. It is also a very good point that it is useful to visualize the results in Table 1, which we do in the revised version along with the table.*

C. Global results. More generally, I think it would be worth reporting globally-averaged values for each variable of interest. It is clear that the authors believe the global trends are positive (decreasing magnitude of ERFaer; e.g., Figure 5). This is also clearly implied by the title. It seems clear that the decreasing aerosol regions dominate in the global average over the increasing region(s), so why not just show this directly?

*Again we agree, although also here there is the "noise" due to the natural-aerosol variability. The numbers are now included in the Table.*

Specific comments:

Line 15: ERFari also includes semi-direct effects.
*Indeed! We thank the reviewer for the clarification and include this in the revision.*

Line 16: If you want a classic reference for ARI as well, I'd recommend Chýlek & Coakley (1974).
*This is a very good suggestion and now included in the revision.*

Lines 22-25: As written, this would imply the world has only warmed ~0.5 K since the pre-industrial, when the true value is closer to 1 K. Instead of just citing $CO_2$ perhaps it'd be better to cite the value for all well-mixed GHGs (sum of ~1.5 K), or state that the aerosol forcing essentially offsets the non-$CO_2$ well-mixed GHG forcing.
*The reviewer again has a very good point. We now clarify the effects of all major anthropogenic forcings.*

Lines 45-47: This sentence could be simplified or broken up. Also, isn't the claim global, not just regional?
*The sentence is revised for readability and removed redundancy. Good point again!*

Line 117: Is significance tested using a t-test? Do you account for temporal autocorrelation?
*Yes, and indeed this information was missing it is now inserted.*

Lines 107-109: Could you provide some more discussion of the differences between the MISR and MODIS trends? Even some statistically significant pixels have opposite trend signs between (a) and (c). Are there differences in the retrievals and their relative strengths/weaknesses or in what conditions retrievals are possible that could help explain this?
*The reviewer indeed has an important point, it is a negligence to not discuss in detail the discrepancy between the MISR and MODIS trend results in particular in the Southern ocean. We do so now and also provide a literature overview over previous studies that already identified (for shorter periods mostly, though) the positive trend in AOD as retrieved from MODIS. One such study proposes it might be due to increases in sea salt emission, but two others find that MODIS stands out with the positive trends: the other products show little or slightly decreasing trends.*

Lines 112-113: If you subset the MODIS and MISR trends to the same period as PMAp, do things look more consistent?
*There are previous studies that look at shorter periods in MODIS as well, and they also show the increasing trends in the Southern ocean. So rather than presenting the new analysis proposed by the reviewer, we now rather report the results of these previous studies.*

Lines 129-130: Especially for LWP, bidirectional changes in response to Nd are now widely acknowledged, so I'm not really sure what the "expected" changes should be in this case.
*The reviewer is right. To make this point clear, we add "not necessarily what is expected".*

Lines 132-133: Similarly, different senses of change in macrophysical cloud properties are possible for different cloud regimes or under different meteorological conditions in the same regime (e.g., Zhang et al., 2022), so it really isn't clear that should be one "expected" change.
*The reviewer is right and this is now explicitly stated at this point of the revised manuscript.*

Line 144: I was a bit surprised by the Gryspeerdt et al. 2016 reference here, as the main point of that paper in my reading is how misleading such correlation analyses can be without the proper statistical controls.
*The reviewer is right, but still Gryspeerdt et al. (2016) concluded that there is a remaining positive relationship between Nd and cloud fraction.*

Lines 158-164: How were model variants treated? Is only one used per model, or do you average all variants for each model, etc.?
*This is an important point! We now clarify that we use the simple arithmetic average.*

Lines 172-173: I'm confused about what the IPCC assessment is referring to here.
Line 174: The emulator ensemble is not introduced.
*We agree that this entire bit on comparison to the IPCC assessment was written in a confusing manner. It is reformulated now: "This result can be compared to the assessment by IPCC AR6 (Forster et al., 2021). Their assessment is based on multiple lines of evidence that are incorporated in an emulator ensemble simulation. The time series of the diagnosed ERFaer is available via the IPCC web site and at https://doi.org/10.5281/zenodo.5705391. Computing the linear trend between 2000 and 2019 yields180 an increase by +0.0145 W m−2 yr−1 between 2000 and 2019 (5 to 95% confidence interval of +0.0068 to +0.0253), i.e. by +0.29 (+0.14 to +0.51) W m−2 over the full period (Gulev et al., 2021; Forster et al., 2021)."*

Figure 4: It might be worth having another figure (perhaps in the supplement) showing each model individually, and perhaps the radiation fields directly (rsutcs, rsut, rsut+rlut) instead of ERF, for a more apples-to-apples comparison with the CERES record.
*Again a very useful suggestion by the reviewer. We split this proposed modification into two. One is to assess the change in radiation vs. the change in aerosol ERF. For this, we now analysed the piClim_histall results*

*that include all changes, and analyse the trends in TOA fluxes. In the solar spectrum, it is evident that the aerosol signal dominates. In the all-sky fluxes, however, a signal by the greenhouse gas forcing is superimposed. This result is reported in the main text.*
*The new Supplementary Figures S6 to S8 now examine the trends for the individual models. No surprises are found, and this result is now also reported in the main text.*

Figure 4: It also may be worth looking at variants versus ensemble average for models like NorESM with several variants to explore how much of the noisiness is due to internal variability.
*This is now also done exemplarily for an individual ESM (NorESM was selected) and shown as Supplementary Figure. The result, namely that the pattern of changes is robust, is reported in the main manuscript.*

Figure 4 caption: The gray shading note is for the wrong figure.
*The reviewer is right, this is corrected now.*

Figure 4: The labels for CERES (rsutcs, etc.) are clear to those familiar with climate modeling but aren't obvious otherwise. Please introduce the labels or use another descriptor.
*This is a good suggestion which we follow in the revision.*

Line 221: More explanation of the Smith et al. (2021a) method would be helpful here, and below for Alright et al. (2021) as well.
*We now added a short paragraph on this in the revised text.*

Line 222: What is the range quoted? I'm guessing 5-95% confidence?
*The reviewer is right and this is now clarified in the text.*

Figure 5: Please explain the colors on the x labels. I think I figured it out after staring at it for a bit, but it would be much easier on readers if the information were in the caption.
*Of course, very good point! It is done now.*

Line 240: Zhou et al. (2021) would also be appropriate to reference here.
*This is an excellent suggestion which we follow.*

Line 245: No strong trends in volcanic aerosol, or eruptions, etc.?
*Indeed, the data shown by Carn et al. (2017) suggest no systematic trends. More detail on their study is now provided in the revised manuscript.*

Lines 245-248: Not only wildfires are relevant here but also agricultural burning, especially in Africa. Andela et al. show that burned area has actually been decreasing on average due to human activities, although there isn't a one-to-one correspondence between burned area and smoke emissions.
*This is a very good point, many thanks to the reviewer for pointing to this reference!*

Table 1: See general comment above, at minimum I would add an "all else" column. I also think it would be helpful to have some indication of how things look in absolute, not relative, units, as spatially averaging the percentage changes doesn't necessarily lead to meaningful values given the differences in the absolute amount of aerosol, etc., involved. For the reported values, are you averaging the percentage values from the maps in the figures in space, or taking the absolute values and calculating the percentage trend for the full region?
*As suggested, a column with the global values is now added. The trends in absolute numbers are now reported as supplementary material.*

Table 2: I believe this table is never introduced?
*The reviewer is right and this mistake is now corrected.We now added a short paragraph on this in the revised text.*

References:

Andela, N., Morton, D., Giglio, L., Chen, Y., van der Werf, G. R., Kasibhatla, P. S., DeFries, R. S., Collatz, G. J., Hantsson, S., Kloster, S., Bachelet, D., Forrest, M., Lasslop, G., Li, F., Mangeon, S., Melton, J. R., Yue, C., and Randerson, J. T.: A human-driven decline in global burned area, Science, 356, 1356-1362, 2017.

Chýlek, P. and Coakley, J. A.: Aerosols and Climate, Science, 183, 75-77, 1974.

Zhang, J., Zhou, X., Goren, T., and Feingold, G.: Albedo susceptibility of northeastern Pacific stratocumulus: the role of covarying meteorological conditions, Atmos. Chem. Phys., 22, 861-880, 10.5194/acp-22-861-2022, 2022.

Zhou, X., Zhang, J., and Feingold, G.: On the Importance of Sea Surface Temperature for Aerosol­Induced Brightening of Marine Clouds and Implications for Cloud Feedback in a Future Warmer Climate, Geophysical Research Letters, 48, e2021GL095896, 10.1029/2021gl095896, 2021.

---

## Author Comment (AC2)

General comment:

This study discusses the evolution of aerosol effective radiative forcing (ERF) in the recent two decades, the period when high quality satellite measurements are available. The authors investigated different aspects of aerosol effects on climate, i.e. aerosol emission, aerosol burden, cloud property, and radiation budget, to assess linear trends of different quantities for these aspects based on both satellite observations and global models. The results show that the observed trends differ in sign on average between regions with negative and positive changes to clear-sky solar ERF in CMIP6 models. Overall, this is a nice overview of recent changes to variables relevant to aerosol effects on climate to identify significant trends for some of them, particularly cloud droplet number concentration and cloud fraction among others. I have relatively minor comments as specified below, and recommend the manuscript be published after the authors address them appropriately.

*On behalf of the co-authors I would like to thank the reviewer for the thorough review of our manuscript.*

Specific comment:

Line 142-143: "In contrast, there are some hints at a change in cloud fraction consistent in pattern and sign with the trends in droplet concentration": Is this derived from Fig.3? Can you provide more specific discussion regarding how cloud fraction trends shown in Fig.3c are interpreted in comparison to droplet concentration in Fig.3a? In general, cloud fraction trends are largely affected by natural meteorological variability, rather than aerosol perturbation, as the authors also pointed out, so it would be very important to demonstrate how aerosol-induced signals can be found in cloud fraction.

*The reviewer is certainly right that there are multiple factors influencing cloud fraction, and aerosols only affect them to a minor, albeit possibly systematic, degree. We now point more clearly to this caveat where we write the statement the reviewer refers to, and also provide more references to the literature that examines at a process level the relationship between aerosols and cloud fraction.*

Line 182-183: "It is split into a strongly decreasing trend in reflected solar radiation and a declining trend in emitted terrestrial radiation (defined positive downwards, so the trend implies more emission to space)": Is the second statement (for terrestrial radiation) in the parentheses correct? I was assuming that the emission to space is decreasing to accelerate global warming (I might be wrong), but if the authors statement is correct, are the two components (solar and terrestrial changes) compensating for each other? I'm a bit confused with the statement here, and would appreciate clarification.

*We agree with the reviewer that this formulation is confusing. We have now formulated the report about the results by Loeb et al. more clearly: "They find this to be due to a strongly decreasing trend in reflected solar radiation, which they attribute to decreased reflection by clouds and sea ice, and a declining trend in emitted terrestrial radiation due to increases in greenhouse gases and water vapour."*

Line 208-211: The CERES data shown in Fig. 4 is discussed only briefly in this short paragraph. Can you provide more detailed discussion on observed radiation trends shown in upper panels of Fig. 4 in more specific comparison to aerosol trends of Figs. 1 and 2 to support the statement of the last sentence?

*The reviewer is right indeed that this discussion needed to be extended. It is done now in the revised version.*

Line 218-219: Can you briefly describe the method of Smith et al. (2021a) to constrain the aerosol ERF by considering the ocean heat uptake?

*We now added a short paragraph on this in the revised text.*

Table 2: Is this referred in the main text? If not, discussing these numbers comprehensively in Section 7 would be beneficial to convey the major message of this study. This table is a very nice summary of the ERFaer change.

*The reviewer indeed points to a negligence! It is now corrected in the revision, where we discuss these outcomes.*

Minor/Editorial point:

Line 172: year -> near (?)

*Well spotted and corrected now!*

Figure 2 caption, line 6: (c) -> (e)
*Again, excellent help by the reviewer!*

---

## Author Comment (AC3)

This manuscript reports the trends of aerosol optical depth, cloud properties, and top-of-atmosphere radiative fluxes in last two decades (2000-2019), mostly from satellite retrievals, to assess the anthropogenic aerosol radiative forcing trends. It also examines the consistency of the trends among AOD, clouds, and radiation. The paper concludes that the anthropogenic aerosol radiative forcing has become globally less negative in this 20-year period, which is consistent with the declining trends of anthropogenic aerosol and precursor emission, aerosol burden, fine-mode aerosols, cloud droplet number concentrations, and TOA fluxes. Based on the findings, it is concluded that the reduction of anthropogenic aerosol leads to an acceleration of the forcing of climate change through both aerosol-radiation and aerosol-cloud interactions.

I find that the manuscript provides an extensive measurement-based information to assess the aerosol radiative forcing on climate, but there are several major issues in synthesize the information to draw the conclusions. Several major issues and specific comments are listed below, and they should be addressed and clarified before the manuscript can be accepted for publication.

*On behalf of the co-authors I would like to thank the reviewer for the thorough review of our manuscript.*

Major Issues:

Definition of ERF: It is not clear what the definition of aerosol ERF is – is it (a) aerosol radiative effects from anthropogenically emitted aerosols and their precursors? Or (b) the ERF from present-day aerosols minus preindustrial aerosols (e.g. 1750)? Or (c) just the radiative effects of total aerosol? Using modern satellite data implies (c), which is present-day total aerosol effects, but in the paper, it is often casually refer that as aerosol climate forcing or anthropogenic aerosol forcing. Clarification is needed.
*The reviewer raises an important point. We always consider both, aerosol-radiation and aerosol-cloud interactions, and as such, a baseline is necessary (no present-day total aerosol effect can be defined for aerosol-cloud interactions, which always require a non-zero baseline). In fact, we have two variants of what the reviewer lists as (b). At some instances, we report the ERF with respect to 1750, in which case we name the baseline. At other instances, we report* changes *in ERF between two time periods, or as trends between these (specifically, for the period 2000 to 2019), in which case no explicit baseline is required. We now clarify this early on in the Introduction: "Throughout this manuscript, we consider ERF with 1750 as baseline, or changes in ERF over certain periods (such as 2000 to 2019)."*

Causality: Even if the trends among aerosol, clouds, and radiative fluxes are "consistent" from satellite observations, it does not mean that the trends can be explained by the reduction of anthropogenic aerosols. These is no effort shown in the paper to separate causality with association. By showing the similarities among the variables is not enough to attribute the trends to the cause. CMIP6 or RFMIP models should be able to provide some insights.
*This is right, and it is especially difficult for the cloud quantities. Unfortunately current GCMs are not very reliable in simulating the cloud response to aerosols, so no clear detection-and-attribution study is possible either. In order to respond to the reviewer concern, we now make it more clear still, in the Conclusions section, that the interpretation of causality is questionable in particular when it comes to cloud fraction trends (cloud liquid water path trends were anyway inconclusive): "It is to be noted that the spatial consistency of the trends in cloudiness are not a clear proof of causality."*

"Consistency" between trends in Fig. 1-4: Global map of trends shown in Figures 1-4 are informative, but more in-depth analysis is needed to not only better convey the consistency (or inconsistency) among the trends of AOD, clouds, and radiative fluxes but also different trends of those quantities in various regions. I would suggest show the 2000-2019 time series of each quality averaged over selected regions (e.g., major pollution source regions, continental outflow regions, and remote regions) to reveal how linear the trends are and if they are indeed consistent with the change of anthropogenic emissions.
*This is indeed a useful suggestion. We did the analysis the reviewer suggested, but (as expected) the time series are (as expected) noisy and far from straight lines. This is now reported in the revised manuscript. We also now make clear what exactly we mean by "consistency" where we first refer to it.*

Significance of the trends: Areas with "substantial" positive and negative trends are defined as those where the clear sky ERF trends are larger than 0.05 W/m2/year from RFMIP multi-model ensemble mean. According to the caption of Table 1, regions with negative trends cover just 7.3% of the Earth's surface and that with positive trends covers 1.1%. That implies no trends or weak trends over 91.6% of the Earth's surface area. How do you explain the significance of global changes of these quantities if the substantial trends are only confined in ~8% of the area?

*The reviewer again highlights one of the key issues. The idea is that trends in forcing are isolated, and not so much trends in natural variability. In order to address the reviewer's remark, we now also report and discuss the global mean trends in Table 1. There is an inconsistency in the AOD trends from MODIS with the other quantities, but overall there is clear support for the main conclusion of reduced aerosol forcing also at the global level. This is now reported in the revised text.*

Specific comments:

Line 8: "consistent" with what? With anthropogenic aerosol trend?
*Indeed this was poorly formulated. We now expand: "cloud droplet numbers show trends in regions with aerosol declines that are consistent with these in sign."*

Line 16: "ERFari occurs through the scattering and absorption of sunlight by aerosols": This is the aerosol radiation interaction, which referred to as "RFari" according to IPCC AR5. The ERFari includes additional "semi-direct effects". Please use the terminology more carefully.
*The reviewer is of course right. We now explicitly mention the semi-direct effect.*

Line 22-25: If +1.01C temperature change is due to CO2 and -0.51C due to aerosol, should the net temperature increase be 1.01 – 0.51 = +0.5C? Or, in other words, it would have reached +1.01C temperature increase without aerosol cooling.
*The reviewer again has a good point! We now clarify also the effect by the all anthropogenic greenhouse gases that allows to better compute the full warming number.*

Line 38: Do you see a turning point from the 20-year data record when aerosol forcing became substantially less negative? That is why plotting time series is very helpful, as I mentioned in "Major issues" #3, to see if the trends are linear or showing a turning point.
*This is a good idea by the reviewer, but the time series are noisy due to lots of interannual variability. It is thus not easy to clearly identify (piecewise) linear trends from such a still rather short period. We will do more in-depth follow-up analysis to seek insights into this point.*

Line 58-59: "anthropogenic aerosol emissions over China have been increasing until ~2010 and decreasing thereafter": That means the emission trend is not only non-linear but also has shifted directions during the past two decades. It will be interesting to see if AOD, clouds, and radiative flux shows similar or different decadal variations.
*The reviewer is right, but for these still shorter periods one would expect the signal to be too noisy to draw firm conclusions. But here again, the reviewer raises an important point which a sub-team (the Univ Leipzig group) will investigate in follow-up work.*

Line 68-69: CMIP6 used the CEDS_v2017 described in Hoesly et al. 2018, not the newest CEDS version (2021 version).
*Indeed this was formulated in a sloppy way and is corrected now.*

Line 74-75: Not very clear what you mean "mirror" here, which usually means opposite direction. Do you mean that OC and BC emissions have an increasing (decreasing) trend that mirrors the decreasing (increasing) trend of sulfur emission in the same region? In Figure 1, the regional trends of SO2, OC, and BC are similar in the same directions, though.
*Sorry for poorness in language. We meant, they show the same. The formulation is now corrected.*

Line 107-108: Over most oceanic area, MODIS and MISR have opposite AOD trends, especially the fine-mode AOD. What is the implication for global aerosol forcing since ocean covers 70% of the surface area?

*First of all, the reviewer has an important point in that it is necessary to discuss this discrepancy in detail. This is now done in the revised manuscript, putting the result reported here into the context of previous studies. Also the relevance for the forcing is now discussed in the new section that reports the global trend numbers.*

Line 111 and 112: Is it Metop-A or Metop-B you are using?
*We are very sorry for the confusion we created. Metop-A is correct and this is rectified now throughout the manuscript.*

Line 114-115: GOME-2 shows different trends over most land regions. Can you be more specific about what the "expected behavior" is? How can the opposite trends in some regions be described as "consistent"?
*This statement explicitly is meant only for the highlighted regions with clear anthropogenic trends. We now clarify that we mean the "same", rather than "expected" behaviour.*

Line 115, "These trends are largely consistent with those from AERONET data": You have not shown any AERONET data here.
*We now report also in the body text at the instance the reviewer highlights that the Aeronet data are shown in Fig. 2.*

Line 131-133: Is CDNC less variable than cloudiness and cloud radiative properties? How large should the variability be to prohibit the detection of trend? Again, time series plots may better convey the story.
*We now make clear that here we mean the variability in cloud fraction and LWP beyond the one driven by aerosols. CDNC responds to aerosols much more directly than these two quantities.*

Line 147-151: There are clearly several regions that the directions or magnitudes of changes between aerosols and cloud properties are not in sync. Can you make more quantitative analysis of different regions, e.g., major pollution regions, immediate downwind regions, and more remote regions, and explain, to the degree you are able, the reason for the consistency or inconsistency between the changes of aerosols and clouds?
*This is indeed an important issue, and perhaps a central reason why it is complicated to infer clues about aerosol-cloud interactions: aerosols are not the first-order determinant on most cloud properties, and so there is not a one-to-one relationship. So for the present analysis, it is proposed to make the link where it is possible (i.e. where there are regionally contingent and consistent changes in the aerosols that may be reflected in changes in cloud properties) in the current study.*

Figure 4: The terms in Figure 4 are confusing. For example, the caption of Fig 4a says "net broadband solar flux for clear-sky", but the figure title indicates it is "rsutcs", which is defined as "radiation shortwave upward TOA clear sky", not "net". Also it seems the quantities from RFMIP in Fig. 4d-f do not corresponding to the quantities from CERES in Fig. 4a-c: CERES data are the radiative fluxes whereas the RFMIP the effective radiative forcing (meaning either PD – PI, or anthropogenic aerosol only). Lastly, rsut + rlut is total (shortwave + longwave), not net. Please get the terms straight and clarify if you are compare the same or different quantities between CERES and RFMIP.
*We clarify now the definitions at the end of the caption and also specify that model and satellite data are comparable. "net" means here, incoming plus reflected (both defined positive downward), and trends for the ERF are shown that are comparable to the trends in ToA net flux changes.*

Figure 4 caption, third line from the bottom, the sentence started with "For the emissions…". What is the context of emission here? Besides, there is no grey shading anywhere in all panels.
*This is a well-spotted mistake that is a remnant after splitting the figures into four. It is now corrected.*

Line 172: Delete "year" in "year0.32 W m-2".
*Corrected.*

Overall, this section is confusing. As I mentioned in "Major Issues" #1, it is not clear whether the discussion is about TOA upward flux, or net flux, or shortwave + longwave flux, or surface downward flux, or if preindustrial condition is considered in the model, or how the RF is defined - is it PD - PI? or is it by anthropogenic aerosol?

*In order to make things very clear, we now start this section with the explanation on what exactly is discussed: "Changes in net top-of-atmosphere radiation fluxes in a period correspond to the changes in ERF in that period, but also include the signal of natural variability and of feedbacks to changing climate. " Also, to further make this clear, we now use the same labels above the panels.*

Line 239: delete "but also".
*Done*

Table 1 caption, 2nd line: Is CMIP6 or RFMIP models used in Fig. 4? Are they the same suite of models? Please be consistent.
*The reviewer is right – it is much better to specifically write RFMIP.*

Line 257-258: CEDS emission was not consider any aerosol satellite retrievals.
*We meant the opposite, namely that in the construction of the emissions inventories, satellite data are considered, but indeed in a mis-leading and too superficial way. It is now clarified.*

Line 269: Remove "aerosol" in "all three aerosol species". SO2 is not aerosol, but an aerosol precursor gas.
*The reviewer is right, this is removed.*

Line 269-270: it is self-repeating that MODIS and MISR AOD increase or decrease at regions aerosols increase or decrease. AOD is a measure of aerosol. Maybe you mean at regions aerosol and precursor emission increase or decrease?
*This is true, corrected.*

Line 272: What are the expectations? Are the expectations consistent with the aerosol trends or not and why?
*The reviewer is right, the formulation was too sloppy. It is now made explicit.*

Line 282-286: Again, it seems the terms you compare between CERES and CMIP6 (or RFMIP?) are not the same terms.
*They are comparable. We now state this at this point of the text: "The changes in net radiation retrieved by CERES should reflect the trends in ERF, but also natural variability and feedbacks to climate change."*

Figures 1-4: Since the color scales are not linear, it is hard to tell the data range covered by the color bars. Please add numbers for each color interval to help quantify the range.
*This is a very good suggestion by the reviewer which we followed.*